# Role of the Gut Microbiome in the Development of Atherosclerotic Cardiovascular Disease

**DOI:** 10.3390/ijms24065420

**Published:** 2023-03-12

**Authors:** Ahmad Al Samarraie, Maxime Pichette, Guy Rousseau

**Affiliations:** 1Internal Medicine Department, Faculty of Medicine, University of Montreal, Montréal, QC H3T 1J4, Canada; 2Cardiology Department, Faculty of Medicine, University of Montreal, Montréal, QC H3T 1J4, Canada; 3Centre de Biomédecine, CIUSSS-NÎM/Hôpital du Sacré-Cœur, Montréal, QC H4J 1C5, Canada

**Keywords:** gut microbiota, gut microbiome, atherosclerotic cardiovascular disease, atherosclerosis, risk factors, hypertension, dyslipidemia, diabetes, trimethylamine N-oxide, secondary bile acids, lipopolysaccharides, short-chain fatty acids

## Abstract

Atherosclerotic cardiovascular disease (ASCVD) is the primary cause of death globally, with nine million deaths directly attributable to ischemic heart diseases in 2020. Since the last few decades, great effort has been put toward primary and secondary prevention strategies through identification and treatment of major cardiovascular risk factors, including hypertension, diabetes, dyslipidemia, smoking, and a sedentary lifestyle. Once labelled “the forgotten organ”, the gut microbiota has recently been rediscovered and has been found to play key functions in the incidence of ASCVD both directly by contributing to the development of atherosclerosis and indirectly by playing a part in the occurrence of fundamental cardiovascular risk factors. Essential gut metabolites, such as trimethylamine N-oxide (TMAO), secondary bile acids, lipopolysaccharides (LPS), and short-chain fatty acids (SCFAs), have been associated with the extent of ischemic heart diseases. This paper reviews the latest data on the impact of the gut microbiome in the incidence of ASCVD.

## 1. Introduction

Despite major advances in prevention and treatment strategies, atherosclerotic cardiovascular disease (ASCVD) remains the leading cause of morbidity and mortality all around the world [1]. This alarming statistic brought to light the complex etiology of atherosclerosis, which has been recognized as not being solely induced by conventional risk factors, such as hypertension, diabetes, dyslipidemia, male sex, and smoking. In 2000, Haraszthy et al. proposed for the first time that the gut microbiota was associated with the occurrence of ASCVD, after finding DNA from multiple bacteria species in a plaque of cholesterol [2].

Trillions of micro-organisms weighing about 1.5 kg inhabit the gut, carrying out key functions that the rest of the human body is incapable of performing [3]. These micro-organisms have combined genomes (the microbiome) that exceed the human genome by many times [4,5,6]. The gut microbiota is dominated by anaerobic bacteria, with *Firmicutes* (Gram-positive) and *Bacteroidetes* (Gram-negative) composing more than 90% of intestinal bacterial species [7].

This paper reviews the latest available information on the role of the gut microbiome in the incidence and progression of ASCVD.

## 2. Metabolic Pathways

Major metabolites have been identified and linked to the development of cardiovascular diseases, including but not limited to trimethylamine N-oxide (TMAO), secondary bile acids, lipopolysaccharides (LPS), short-chain fatty acids (SCFAs), and phenylacetylglutamine (PAGln). Figure 1 illustrates major gut metabolic pathways leading to ASCVD.

First, TMAO directly contributes to the pathogenesis and extent of ASCVD. The correlation between these two entities was first reported by Wang et al. in 2011 [8]. TMAO goes through several enzyme modifications before being transformed into its final active form. In fact, the first step requires the intestinal conversion of one of three metabolites found in the food, namely L-carnitine, choline, and betaine, into TMA by the enzyme TMA lyase, which is derived from the *Firmicutes* species found in the gut microbiota [9]. These three nutrients are naturally found in foods, such as eggs, red meat, and fish. After their conversion into TMA, this amine is absorbed into the bloodstream before being transported to the liver, where it is transformed into TMAO by the enzyme flavin-dependent mono oxygenase 3 (FMO_3_) [10]. In physiologic states, the kidney excretes in the urine close to 95% of TMA oxidized into TMAO [11]. Hence, changes to any component along this metabolic pathway, from food ingestion to hepatorenal function together with the liver FMO_3_ activity, could lead to increased levels of TMAO with associated complications, such as ASCVD [12].

Primary bile acids are synthesized in the liver from cholesterol and are conjugated with glycine, subsequently leading to the formation of cholic acid and chenodeoxycholic acid. Then, primary bile acids are transported in the gut, in which the microbiota contributes to their deconjugation to form secondary bile acids in the distal ileum [13]. Secondary bile acids permit absorption of lipid nutrients and fat-soluble vitamins [14]. They are also involved in the activation of two key receptors, namely farnesoid X receptor (FXR) and takeda G protein-coupled receptor 5 (TGR5). These receptors modulate glucose and cholesterol metabolism. In fact, TGR5 leads to an increased secretion of glucagon-like peptide 1 (GLP-1), which contributes to an improved glucose tolerance [15,16]. TGR5 is also thought to possess anti-inflammatory properties by inhibiting nuclear factor-κB (NF-κB), thus decreasing the production of pro-inflammatory cytokines [17]. In another trial, simultaneous inhibition of both FXR and TGR5 exacerbated atherosclerotic formation, thereby highlighting their benefits in disease control [18]. Their anti-inflammatory and anti-atherogenic properties arise from suppression of tumor necrosis factor-α (TNF-α) and NF-κB signaling pathways in addition to a decreased secretion of pro-inflammatory cytokines [19]. Therefore, secondary bile acids activate two key receptors involved in the inhibition of major atherosclerotic pathways.

Similar to TMAO, LPS are endotoxins found on the outer membrane of Gram-negative bacteria and are involved in the pathogenesis of ASCVD. LPS are recognized by the innate immune system by toll-like receptor 4 (TLR4), which is a subtype of pattern recognition receptors (PRR) [20]. Upon recognition of LPS, TLR4 induces a pro-inflammatory state with an increased production and secretion of cytokines and chemokines [21]. LPS are also identified by other receptors, such as LPS-binding protein (LBP), myeloid differentiation protein 2 (MD-2), and cluster of differentiation 14 (CD14) [22]. These receptors, which are mainly expressed on macrophages, activate and enhance several protein kinases, such as IL-1 receptor-associated kinase (IRAK-1) and myeloid differentiation factor 88 (MyD88). NF-κB is subsequently activated, which, along with LPS, stimulate numerous pro-atherosclerotic inflammatory pathways [23,24,25]. In fact, LPS induce endothelial dysfunction, increase oxidative stress through production of reactive oxygen species (ROS), and produce several pro-inflammatory cytokines, such as TNF-α, interleukin-1 (IL-1), interleukin-6 (IL-6), and interleukin-8 (IL-8) [26,27,28]. Therefore, LPS contribute to ASCVD by promoting inflammation through various pathways.

In contrast to TMAO and LPS, SCFAs are protective against the occurrence of atherosclerosis and are the result of the ingestion and digestion of complex carbohydrates by numerous gut bacteria, including *Anaerostipes butyraticus*, *Faecalibacterium prausnitzii*, and *Roseburia intestinalis* [29,30]. The most frequent SCFAs produced are acetate, butyrate, and propionate [31]. These nutrients serve many roles, but their primary function is to modulate the host immune system through increased production of regulatory T cells and suppression of histone deacetylases (HDACs) [32,33]. By inhibiting HDACs, SCFAs inhibit inflammatory pathways owing to a decrease in NF-κB activation together with a reduced production of pro-inflammatory cytokines [34]. Other functions attributed to SCFAs include enhanced intestinal barrier stability and protection against pathogen invasion [32,33]. Thus, SCFAs protect against atherosclerosis by modulating inflammatory pathways.

Phenylacetylglutamine (PAGln) is a metabolite that was recently discovered and was shown to be positively associated with the development of cardiovascular diseases [35]. PAGln is derived from a simple amino acid, phenylalanine, that undergoes a series of alterations before arriving to its active metabolite. In fact, Nemet et al. have demonstrated that the microbial *porA* gene permits the transformation of phenylalanine into phenylacetic acid, with subsequent hepatic metabolization of phenylacetic acid into PAGln [36]. PAGln was first reported to be positively correlated with ASCVD and overall mortality in patients suffering from chronic kidney disease (CKD) [37]. Several pathophysiologic mechanisms were hypothesized to explain this association. As a matter of fact, PAGln was shown to increase platelets’ activation and responsiveness, resulting in increased thrombosis potential leading to ASCVD. [38] PAGln also transmits cellular events via G-protein coupled receptors, specifically the α2A, α2B, and β2 adrenergic receptors [36]. Interestingly, carvedilol, a commonly used β-blocker in clinical practice, was shown to inhibit these prothrombotic effects [36]. Thus, PAGln is involved in the occurrence of ASCVD through an accelerated rate of thrombus generation and vessel occlusion, potentially giving rise to acute myocardial infarction.

## 3. Gut Microbiome and Hypertension

Arterial hypertension is a well-recognized and major risk factor for the development of ASCVD [39,40,41]. Leading medical communities of cardiology recommend blood pressure control both pharmacologically and non-pharmacologically as part of cardiovascular diseases’ primary and secondary preventions [42,43,44]. Even though the exact cause of essential hypertension remains unclear, many risk factors are thought to contribute to its development, including advanced age, positive family history, obesity, a high-sodium diet, and a sedentary lifestyle [42,43,44].

Recently, the gut microbiota has been found to play a role in the development of hypertension [45,46,47,48,49,50,51,52,53,54]. Li et al. have demonstrated that high blood pressure was transferrable through fecal transplantation from hypertensive subjects to germ-free mice, thus confirming the implication of the intestinal microbiome [47]. Additionally, compared to healthy controls, pre-hypertensive and hypertensive individuals have decreased microbial richness and variety with an overgrowth of specific bacteria, namely *Prevotella* and *Klebsiella* [47]. A decrease in microbial richness constitutes an alteration in gut microbiome, thus defining dysbiosis. Dysbiosis is thought to induce low-grade inflammation, which in turn can provoke hypertension when the inflammation is persistent [55,56]. Additionally, a reduction in *Lactobacillus* abundance can induce higher blood pressure values in both mice and humans when compared to healthy controls [57].

Yang et al. recently proved that the *Firmicutes* on *Bacteroidetes* ratio was increased in spontaneously hypertensive rats, in angiotensin II-induced hypertensive rats, and in a small group of humans with hypertension [51]. It is noteworthy to note that by normalizing this ratio with the administration of minocycline, blood pressure of spontaneously and induced-hypertensive rats also normalizes [51]. Additionally, fasting for a five-day period seems to induce a modification in the gut microbiota, subsequently reducing blood pressure in hypertensive patients [58].

The gut microbiota produces various metabolites with different effects on blood pressure regulation [59]. Beneficial metabolites include SCFAs and vitamins. Acetate, propionate, and butyrate account for 80% of the total SCFAs produced [60]. SCFAs are thought to be beneficial in blood pressure reduction through mainly their vasorelaxant and anti-inflammatory effects [61]. Indeed, Bartolomaeus et al. demonstrated that the administration of propionate in mouse models was associated with a better control of high blood pressure together with a decrease in vascular inflammation and cardiac damage [62]. In another mouse model, acetate was shown to be highly effective in improving cardiac function by reducing left ventricular wall thickness and body weight in addition to a decrease in systemic blood pressure [63].

By contrast, TMAO, another metabolite produced by the gut microbiome, is positively associated with hypertension [64]. TMAO has a proatherogenic and prothrombotic effect [65,66]. This toxic metabolite is thought to induce hypertension through prolongation of the hypertensive effect of angiotensin II and facilitation of angiotensin II-induced vasoconstriction [67,68]. TMAO also enhances stiffening of the large arteries, namely the aorta and carotid arteries, which amplifies the risk of ASCVD both directly and indirectly through increased systolic blood pressure [69].

Thus, the gut microbiome fabricates different metabolites with various effects on blood pressure. SCFAs improve its control while TMAO is deleterious.

## 4. Gut Microbiome and Diabetes

In 2019, diabetes was estimated to affect around 463 million people worldwide [70]. That alarming number is expected to rise to 700 million by 2045 [70]. Obesity contributes to the development of type 2 diabetes (T2D) through many pathophysiologic mechanisms, mainly insulin resistance [71]. Diabetes results in microvascular and macrovascular complications, with cardiovascular disease being the most common cause of morbidity and mortality among people suffering from diabetes [72]. Like hypertension, the exact etiology of diabetes remains unclear but many risk factors have been identified, including a positive family history, advanced age, obesity, hypertension, and a history of cardiovascular disease [73,74].

In 2004, Backhed et al. suggested for the first time that the gut microbiota could be linked to the development of T2D by inducing alterations in glucose metabolism [75]. Many studies have reported that obesity and alterations in glucose metabolism were associated with an altered ratio between the two most common bacteria composing the intestine, with increased *Bacteroidetes* and decreased *Firmicutes* levels [76,77,78]. Thereafter, experiments using metagenomic sequencing in human volunteers established that people with T2D have a dysbiotic gut microbiota [7,79]. Both trials reported that people with diabetes had less butyrate-producing bacteria. Butyrate is one of the three main SCFAs and is thought to possess an advantageous effect on insulin sensitivity and energy balance [80]. Several studies have followed and have reported that gut microbiome dysbiosis contributes to a less favorable course of T2D by inducing a rapid progression of insulin resistance [81,82,83,84,85].

More than 80% of patients with T2D are overweight [86]. The underlying pathophysiological mechanism linking these two conditions is insulin resistance induced by obesity [87]. Numerous studies from animal models have demonstrated that the gut microbiota is implicated in the development of obesity [75,76,77,88,89]. Several trials have provided an explanation, with one interesting experiment reporting that low bacterial variety in the microbiome is associated with insulin resistance, fatty liver, low-grade inflammation, and obesity when compared to high bacterial diversity [90]. In another experimentation, scientists isolated the microbiota of obese animals and transplanted it into germ-free animals; obesity developed after 14 days [75]. Other trials have focused on the potential role of SCFAs and have found that mice suffering from diabetes exhibit lower levels of butyrate-producing bacteria, such as *Fecalibacterium prausnitzii*, *Eubacterium rectale*, and *Roseburia intestinalis*, when compared to healthy controls [91,92]. Thus, SCFAs, particularly butyrate, are beneficial metabolites that seem to protect against the incidence of diabetes.

In addition, LPS have been found to be early triggers of obesity by inducing an inflammatory state through secretion of cytokines and chemokines [93]. In healthy individuals, the ingestion of a high-fat meal leads to a transitory increase in plasma LPS levels while in patients suffering from obesity and insulin resistance, LPS levels were found to be chronically elevated, thus contributing to the development of T2D [94,95]. 

Furthermore, recent animal studies have suggested that elevated levels of circulating TMAO are associated with an increased risk of developing T2D, mainly through impaired glucose tolerance, insulin resistance, and oxidative stress [96,97]. Chronic high levels of TMAO are also linked with an increased risk of obesity via secretion of inflammatory cytokines, thus contributing to the occurrence of T2D [98]. A recently published meta-analysis confirmed previous findings and suggested a positive association between T2D and TMAO levels in a dose-dependent manner [99].

Gut microbiota is strongly linked with both microvascular and macrovascular diabetic complications [100,101,102,103]. Indeed, patients with end-stage diabetic nephropathy were found to have an abundance of *Haemophilus* and *Lachnospiraceae* bacteria when compared to earlier stages [104]. Furthermore, *Pasteurellaceae* bacteria are significantly lower in patients with diabetic retinopathy as compared to patients without this complication [105]. Plasma TMAO levels are also significantly increased in individuals with diabetic retinopathy, and its levels are associated with the incidence of this microvascular complication [106,107]. These data highlight the importance of the intestinal microbiome and suggest that dysbiosis could play an important role in the development of diabetic complications.

## 5. Gut Microbiome and Dyslipidemia

Dyslipidemia is one of the major risk factors for both the occurrence and progression of cardiac diseases [108]. In recent decades, primary and secondary prevention strategies were implemented to decrease cholesterol levels. Despite major improvements, dyslipidemia still affects around 12% of adults in the United States [109]. Some of the identified risk factors for increased cholesterol levels are obesity, lack of physical activity, smoking, an unhealthy diet, and diabetes [110,111,112]. Uncontrolled diabetes is one of the most common conditions contributing to dyslipidemia through insulin resistance, therefore leading to hyperinsulinemia. Elevated insulin concentrations contribute to increases in both low-density lipoprotein-cholesterol (LDL-C) and triglycerides levels in contrast to fewer high-density lipoprotein-cholesterol (HDL-C) particles [113,114,115].

Gut microbiota has been shown to be involved in the occurrence of hyperlipidemia [116]. In fact, a recent study reported that people with dyslipidemia exhibit lower levels of fecal butyrate, acetate, and propionate when compared to healthy controls [117]. These metabolites represent the main SCFAs and are produced by a variety of gut bacteria, such as *Bifidobacterium*, *Lactobacillus*, *Faecalibacterium prausnitzii*, and *Roseburia* [118]. Indeed, they are thought to protect against obesity and diabetes by improving lipid and glucose homeostasis as well as glucose tolerance [80,119,120,121].

Additionally, in contrast to a lower abundance of SCFAs, patients with high levels of cholesterol excrete feces with a higher quantity of LPS-producing bacteria, such as *Escherichia coli* and *Enterobacter cloacae* [118]. LPS compose the cell walls of Gram-negative bacteria and are responsible for the release of pro-inflammatory cytokines [122]. An overproduction of these cytokines leads to increased circulating levels of nitric oxide, subsequently triggering a global activation of inflammatory reactions resulting in cardiac, renal, hepatic, and pulmonary failures [123,124,125].

Furthermore, patients with dyslipidemia tend to exhibit high levels of TMAO, which reduce levels of HDL-C, therefore increasing the risk of ischemic heart disease [126,127]. TMAO was also shown to reduce the expression of cytochrome P450 family 7, subfamily A member 1 (CYP7A1), which is a key enzyme in cholesterol and bile acid metabolism, in addition to inhibiting cholesterol transport, thus inducing cholesterol accumulation in cells [128]. 

Finally, the gut microbiome is involved in the production of secondary bile acids, which were shown to be protective against the development of dyslipidemia [129,130]. In fact, these metabolites modulate glucose and cholesterol metabolism through the activation of two key receptors, specifically FXR and TGR5 [131,132]. Several studies have established that a deficiency in any one of these receptors, particularly FXR, leads to dyslipidemia, with increased triglycerides and non-HDL-C levels [133,134,135]. In contrast, the activation of FXR by secondary bile acids increases the activity and expression of LDL receptors in addition to an inhibition of the activity of proprotein convertase subtilisin/kexin type 9 (PCSK9) [136,137,138]. Thus, FXR activation by the gut microbiome could lower LDL-C levels and contribute to a better control of dyslipidemia. 

Altogether, the previous data suggest that SCFAs and secondary bile acids are protective against the incidence of dyslipidemia while other metabolites, such as LPS and TMAO, are detrimental, contributing to an increase in cholesterol levels.

## 6. Gut Microbiome and Atherosclerotic Cardiovascular Disease

Even though there has been substantial improvement in cardiovascular disease outcomes in the past few decades, ASCVD remains the leading cause of death around the world [139,140,141]. Insufficient prevention strategies and uncontrolled risk factors are the reasons why cardiac diseases still top the list [139,140]. Uncontrolled dyslipidemia, persistent inflammation, and high levels of oxidative stress greatly contribute to atherosclerosis [142].

Traditionally, ASCVD prevention strategies focused solely on lifestyle modifications, such as eating a healthy diet and doing exercise, in addition to taking beneficial medications, such as aspirin and beta-blockers [143,144,145]. The gut microbiota was neglected until the scientific community realized its importance in playing key functions in the body, thus labeling it “the forgotten organ” [146]. At the beginning of the millennium, a relationship between the microbiota and atherosclerosis was demonstrated for the first time, after multiple studies reported the presence of DNA of numerous bacterial species in a plaque of cholesterol [2,147]. Another trial suggested the presence of dysbiosis in individuals suffering from atherosclerosis, with an abundance of *Actinobacteria* in their intestine as compared to a large quantity of butyrate-producing bacteria in healthy controls [148].

Recent studies in mice have suggested an important role of the gut microbiota in converting dietary phosphatidylcholine to TMA, which is then oxidized in the liver to TMAO [8,149]. TMAO is a pro-atherosclerotic molecule, with patients suffering from ASCVD displaying significantly higher levels of TMAO as compared with healthy individuals [149]. Several experiments have demonstrated that TMAO was involved in all steps leading to atherogenesis, specifically foam cells’ formations, endothelial dysfunction, thrombus generation, and plaque instability leading, ultimately, to plaque rupture and acute coronary syndrome [8,65,66,150].

Five mechanisms linking TMAO to ASCVD have been recently proposed. Figure 2 summarizes these mechanisms. The first pro-atherosclerotic mechanism is an increased migration of macrophages and an augmented formation of foam cells in cholesterol plaques [151,152]. In a trial involving mice supplemented with either choline or TMAO, Park et al. found that scavenger receptor-A (SR-A) and CD36, two macrophage receptors associated with atherosclerosis, were both increased when compared to control mice [153].

The second metabolic pathway states that TMAO impacts cholesterol metabolism by inhibiting the reverse cholesterol transport (RCT) system and diminishing cholesterol excretion through the biliary system [66]. The RCT system helps maintain cholesterol homeostasis by transporting cholesterol from peripheral tissues to the liver for biliary excretion [154]. A recently published study demonstrated a 35% decrease in the RCT system in mice fed with a diet containing TMAO when compared with healthy controls [66]. TMAO also increases cholesterol levels by downregulating two cytochromes, namely CYP7A1 and CYP27A1. Downregulation of these enzymes leads to decreased bile acid secretion, which reduces cholesterol excretion, thereby contributing to accelerated atherosclerosis [155,156].

The third likely pathway is TMAO-induced endothelial dysfunction [157]. Indeed, TMAO induces vascular inflammation by increasing recruitment of leucocytes to endothelial cells through a G-protein-coupled receptor (GPCR) pathway [158]. TMAO also causes inflammation through mitogen-activated protein kinase (MAPK) and NF-κB signaling pathways [158]. Finally, protein kinase C is a known mediator of endothelial dysfunction, and its activity was found to be significantly increased in response to a diet enhanced with TMAO [159,160].

The fourth pathway involves an increase in oxidative stress. Recent trials have demonstrated that TMAO activates the nucleotide-binding oligomerization domain-like receptor family pyrin domain-containing 3 (NLRP3) inflammasome in endothelial cells. Activation of the NLRP3 inflammasome is involved in the production of ROS through activation of the mitochondrial reactive oxygen species signaling pathway [161,162]. Oxidative stress causes cell damage and is involved in the pathogenesis of multiple diseases, including ASCVD [163].

The fifth and last identified pathway mediates TMAO-associated atherosclerosis through suppression of endothelial progenitor cells’ (EPC) production [164]. Several trials have demonstrated that decreased levels of EPCs contribute to endothelial dysfunction, since EPCs are known for their role in repairing and regenerating damaged endothelium following vascular injury [165,166,167]. Chou et al. have found that TMAO levels were proportional to plasmatic inflammatory markers, specifically high-sensitivity C-reactive protein (hsCRP), IL-6, and TNF-α [164]. By contrast, TMAO levels are inversely proportional to EPC levels, thus leading to impaired endothelial function [164].

Other than TMAO, LPS are other pro-atherosclerotic metabolites released by Gram-negative bacteria [168]. In healthy individuals, butyrate is secreted by the gut microbiota in a sufficient amount to maintain the intestinal barrier [169]. In atherosclerosis, gut microbiome dysbiosis results in a reduced number of butyrate-producing bacteria, subsequently leading to increased intestinal permeability and increased LPS levels [170,171]. LPS activate numerous inflammatory pathways that contribute to the occurrence of atherosclerosis. Indeed, LPS induce the generation of ROS by activating nicotinamide adenine dinucleotide phosphate (NADPH) oxidase [172]. NADPH oxidase produces ROS and induces the production of pro-inflammatory cytokines, such as TNF-α, IL-6, and IL-8 [173]. Furthermore, LPS provoke expression of inflammatory mediators, resulting in increased infiltration of inflammatory cells in cholesterol plaques [174]. These inflammatory cells include neutrophils, monocytes, selectins, and integrins, and are involved in the progression of atherosclerosis [175]. Thus, LPS directly contribute to the development and progression of atherosclerosis. 

Moreover, the gut microbiome produces secondary bile acids, which are involved in the activation of two key receptors, the membranous TGR5 and the nuclear FXR [130]. Their activation is associated with a slowed progression of atherosclerosis through an inhibition of NF-κB activity, resulting in a decreased production of pro-inflammatory cytokines, as well as inhibited LDL uptake via a lowered expression of CD36 [130]. In contrast, the absence of FXR in mouse models was demonstrated to be linked with a decreased survival rate owing to more severe atherosclerosis with increased atherosclerotic plaque burden [176].

Likewise, SCFAs are thought to be beneficial on ASCVD by inhibiting various inflammatory mechanisms thought to induce atherosclerosis. SCFAs are produced by the gut microbiota through fermentation of dietary fibers [60,177]. Ingestion of a high-fiber diet contributes to an improved glycemic control and weight loss as well as increased blood concentrations of SCFAs [178,179,180]. SCFAs, particularly butyrate, were recently shown to suppress atherosclerotic lesions in mice supplemented with a high-fiber diet [181,182]. Butyrate is thought to increase plaque stability by decreasing ROS and nitric oxide release from macrophages as well as reducing production of known inflammatory molecules, such as chemotaxis protein-1, vascular cell adhesion molecule-1, and matrix metalloproteinase-2 [181,182].

A recently published trial demonstrated that a dysbiotic microbiota is positively associated with an increase in the size of acute myocardial infarction in rats [183]. Probiotics were shown to attenuate the infarct size observed in the dysbiotic group suggesting that microbiota is an important component of ischemic damage. In addition to an increase in infarct size, other noteworthy findings include a higher plasma LPS concentration secondary to increased gut permeability together with an increased *Firmicutes* to *Bacteroidetes* ratio.

Thus, dysbiosis contributes to the development of atherosclerosis through increases in TMAO and LPS levels while secondary bile acids and SCFAs are protective. Figure 3 illustrates the implication of the gut microbiome in the occurrence of ASCVD.

## 7. Conclusions

In the last few decades, major advances have been made in the understanding of physiological and pathological functions of the gut microbiota. In the cardiovascular field, there is no doubt nowadays that the microbiome plays a crucial role in the development of ASCVD. The microbiome is directly involved in all steps leading to atherogenesis, including all major cardiovascular risk factors, specifically hypertension, obesity, diabetes, and dyslipidemia. In the upcoming years, the challenge will be to transition from theoretical understanding to clinical practice as major pathophysiologic mechanisms linking the gut microbiota to ASCVD have been elucidated. In the near future, can an in-depth analysis of the gut microbiota be used as a cardiovascular risk marker for which the use of probiotics might prove beneficial when used adequately?

## Figures and Tables

**Figure 1 ijms-24-05420-f001:**
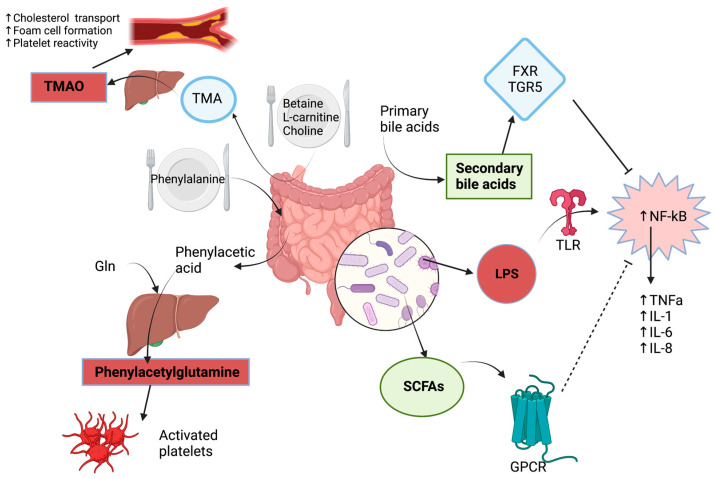
Major metabolic pathways involving the gut microbiome and leading to the development and progression of atherosclerotic cardiovascular disease. FXR: farnesoid X receptor, Gln: glutamine, IL-1: interleukin-1, IL-6: interleukin-6, IL-8: interleukin-8, LPS: lipopolysaccharides, NF-κB: nuclear factor-κB, SCFAs: short-chain fatty acids, TGR5: takeda G protein-coupled receptor 5, TLR: toll-like receptor, TMA: trimethylamine, TMAO: trimethylamine N-oxide, TNF-α: tumor necrosis factor-α.

**Figure 2 ijms-24-05420-f002:**
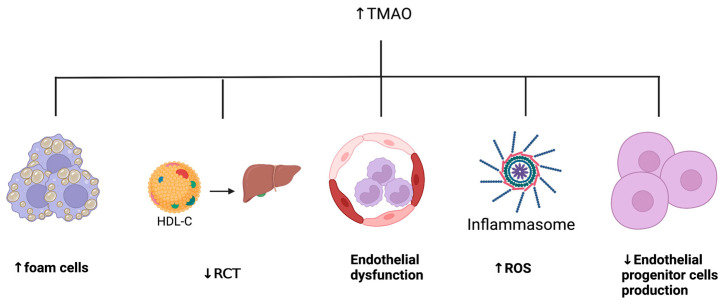
Proposed mechanisms linking trimethylamine N-oxide to atherosclerotic cardiovascular disease. HDL-C: high-density lipoprotein-cholesterol, RCT: reverse cholesterol transport, ROS: reactive oxygen species, TMAO: trimethylamine N-oxide.

**Figure 3 ijms-24-05420-f003:**
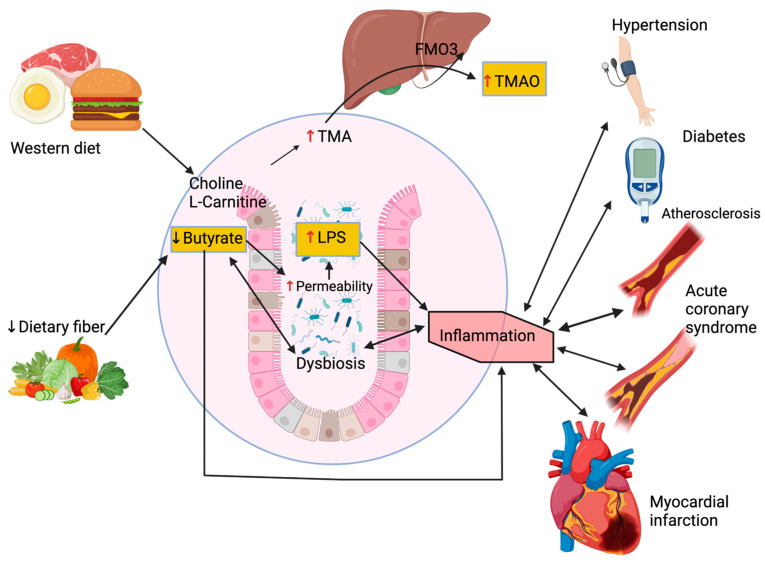
Role of the gut microbiome in the incidence of atherosclerotic cardiovascular disease. A low-fiber diet is associated with a decreased production of the short-chain fatty acid butyrate, subsequently aggravating dysbiosis as well as sustaining local and systemic inflammation through leakage of bacterial toxins, notably LPS. A modern western diet rich in red meat promotes bacterial production of TMA, which is then oxidized to the pro-atherosclerotic metabolite TMAO in the liver. FMO3: flavin-containing monooxygenase 3, LPS: lipopolysaccharides, TMA: trimethylamine, TMAO: trimethylamine N-oxide.

## Data Availability

No new data were created or analyzed in this study. Data sharing is not applicable to this article.

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
