# Peer review of "Role of the Gut Microbiome in the Development of Atherosclerotic Cardiovascular Disease"

_ijms, 2023, doi:10.3390/ijms24065420_

Round 1
Reviewer 1 Report
This manuscript is the research on the " Role of the microbiome in the development of atherosclerotic cardiovascular disease " submitted by Ahmad Al Samarraie, Maxime Pichette, and Guy Rousseau. The manuscript has some points that should be improved.
1. The abstract should be highlighted more.
2. The involvement of primary and secondary bile acids in ASCVD metabolism is suggested for more discussion in the article.
3. Phenylacetylglutamine (PAGln) has only one piece of literature, and the presentation of the evidence needs to be revised and improved.
4. What is the effect of the increase or decrease of SCFAs on ASCVD? Please add more information.
5. Line 212 What kind of potential probiotics?
6. Line 231 Plasma TMAO levels are also significantly increased in people with diabetes with retinopathy; what is the mechanism?
Author Response
We would like to start by thanking both reviewers for their comments. They helped us greatly improve our paper. Here is our response to their comments:
Reviewer 1
- The abstract should be highlighted more.
The abstract has been revised and updated to highlight the importance of the gut microbiota and its metabolites in the incidence of ASCVD.
- The involvement of primary and secondary bile acids in ASCVD metabolism is suggested for more discussion in the article.
The involvement of bile acids in ASCVD metabolism is discussed throughout the article. We have detailed how they are formed in the second section “Metabolic Pathways” and how they inhibit major atherosclerotic pathways. We have then discussed in the fifth and sixth sections how secondary bile acids are protective against the occurrence of dyslipidemia and ASCVD, mainly through the activation of FXR and TGR5.
- Phenylacetylglutamine (PAGln) has only one piece of literature, and the presentation of the evidence needs to be revised and improved.
Thank you for this comment. To our knowledge, there are not many papers describing the association between PAGln and ASCVD. We have reviewed the literature and we were able to find three additional relevant manuscripts. We were therefore able to improve the paragraph about PAGln.
- What is the effect of the increase or decrease of SCFAs on ASCVD? Please add more information.
As described in the paper, SCFAs are protective against ASCVD by inhibiting pro-atherosclerotic inflammatory pathways. They are also protective against the occurrence of all major ASCVD risk factors, specifically hypertension, diabetes, and dyslipidemia, through various mechanisms described in the article in their specific section.
- Line 212 What kind of potential probiotics?
The word “probiotics” was used instead of “metabolites”. We have made the correction. Thank you.
- Line 231 Plasma TMAO levels are also significantly increased in people with diabetes with retinopathy; what is the mechanism?
Thank you for this question. We have reviewed the literature and have found only two papers that explore the association between TMAO and diabetic retinopathy. No mechanism is provided. We assume it is probably the same mechanism as the one by which TMAO induces diabetes. However, to this day, our assumption was never confirmed by a published piece of literature.
Reviewer 2 Report
To the authors
This manuscript reviewed the role of gut microbiota in atherosclerotic diseases. They summarized the way that gut microbiome affect the pathophysiology of atherosclerosis directly and through its risk factors. I think that the readers will be interested in this article. There are several comments.
1. The authors mentioned the metabolites of gut microbiota, such as TMAO, secondary bile acids, and LPS in the two different parts. I feel that slightly redundant. If possible, I think that the authors can restructure the text.
2. The authors used the word “diabetics” several times. Recently, using this word is thought to be avoided. The authors should use person-centered language.
3. The word “probiotic” is defined as a food or pill that contains good bacteria that may keep you healthy. In this manuscript, the authors used probiotics for metabolites of bacteria. I felt slightly uneasy with this word. Please consider the use of this word.
4. In line 246: Do LDL levels increase, or HDL and triglycerides levels?
5. Line 318: “CD” has been described before.
6. Figure 3: Dietary fiber was not mentioned in the main text. Please describe and discuss the role of dietary fiber in the main text.
7. Figure 3: I felt that this figure represented intestinal villi, but the space under the mucosa showed intestinal lumen. It seems slightly confusing. Please change it easy to understand.
Author Response
We would like to start by thanking both reviewers for their comments. They helped us greatly improve our paper. Here is our response to their comments:
Reviewer 2
- The authors mentioned the metabolites of gut microbiota, such as TMAO, secondary bile acids, and LPS in the two different parts. I feel that slightly redundant. If possible, I think that the authors can restructure the text.
The metabolites are mentioned in both sections on purpose. In the “Metabolic Pathways” section, we explore how these gut metabolites arise and how they contribute to atherosclerosis. In the following sections, all metabolites are mentioned again, but this time, we explore mechanisms by which they give rise to ASCVD major risk factors, namely hypertension, diabetes, and dyslipidemia. Some of the mechanisms are the same, which is why they can be found in both parts.
- The authors used the word “diabetics” several times. Recently, using this word is thought to be avoided. The authors should use person-centered language.
Corrections have been made. Thank you.
- The word “probiotic” is defined as a food or pill that contains good bacteria that may keep you healthy. In this manuscript, the authors used probiotics for metabolites of bacteria. I felt slightly uneasy with this word. Please consider the use of this word.
Corrections have been made. Thank you.
- In line 246: Do LDL levels increase, or HDL and triglycerides levels?
LDL and triglycerides levels increase while HDL levels decrease.
- Line 318: “CD” has been described before.
We have made the correction. Thank you.
- Figure 3: Dietary fiber was not mentioned in the main text. Please describe and discuss the role of dietary fiber in the main text.
We added some text about the mechanism by which dietary fibers contribute to the production of SCFAs. We also highlighted their beneficial role on glycemic control and weight loss.
- Figure 3: I felt that this figure represented intestinal villi, but the space under the mucosa showed intestinal lumen. It seems slightly confusing. Please change it easy to understand.
We have made the correction. Thank you.
Round 2
Reviewer 2 Report
The manuscript is improved in its revised version.